# A Limited Course of Eculizumab in a Child with the Atypical Hemolytic Uremic Syndrome and Pre-B Acute Lymphoblastic Leukemia on Maintenance Therapy: Case Report and Literature Review

**DOI:** 10.3390/jcm11102779

**Published:** 2022-05-14

**Authors:** Daniel Turudic, Danko Milosevic, Katarina Bilic, Zoltán Prohászka, Ernest Bilic

**Affiliations:** 1Department of Pediatric Hematology and Oncology, University Hospital Centre Zagreb, Kispaticeva 12, 10000 Zagreb, Croatia; danielturudic@gmail.com (D.T.); ernestbilic@gmail.com (E.B.); 2School of Medicine, University of Zagreb, Šalata 3, 10000 Zagreb, Croatia; katarina91298@gmail.com; 3Department of Pediatrics, General Hospital Zabok and Hospital of Croatian Veterans, Bracak 8, 49210 Bracak, Croatia; 4Department of Internal Medicine and Haematology, Semmelweis University, 1085 Budapest, Hungary; prohaszka.zoltan@med.semmelweis-univ.hu; 5Research Group for Immunology and Haematology, Semmelweis University, 1085 Budapest, Hungary

**Keywords:** atypical hemolytic uremic syndrome, acute lymphoblastic leukemia, 6-mercaptopurine, drug-induced, child

## Abstract

Acute lymphoblastic leukemia (ALL) is considered a possible risk for the occurrence of thrombotic microangiopathies. We present a girl with pre-B ALL successfully treated according to the BFM ALL IC-2009 protocol on maintenance therapy followed by aHUS occurrence. This is the seventh case of HUS/aHUS on ALL maintenance therapy and the first with clearly documented eculizumab use in the early stage of aHUS/secondary TMA. Standard and additional parameters were used in aHUS monitoring alongside the reticulocyte production index adjusted for age (RPI/A) and the aspartate aminotransferase-to-platelet ratio index (APRI) as markers of hemolysis and rapid response following treatment. RPI/A and APRI are markers of bone marrow response to anemia serving as red blood cell vs. platelet recovery markers. Together they mark the exact recovery point of thrombotic microangiopathy and serve as a prognostic marker of eculizumab treatment success. During the 8-month treatment and 6-month follow-up, no recurrence of hemolysis, ALL relapse, or renal damage were observed. A systematic review of the literature revealed 14/312 articles; five children had aHUS before the onset of ALL, and two children had both diseases concurrently. At least 3/7 patients are attributed to aHUS, of whom 2/7 have renal damage. Potential undiagnosed/unpublished cases may be assumed.

## 1. Introduction

Increased risk for acquiring thrombotic microangiopathy (TMA) in malignant diseases is described primarily in adult patients [1,2]. According to opinions from some previous authors, leukemia in children, per se, represents a rare but possible risk for the occurrence of a typical/atypical hemolytic uremic syndrome (HUS/aHUS) [3,4,5,6,7,8,9,10,11,12,13,14,15]. This conclusion is attributed to the genetic background of these hematologic diseases that have the potential for aHUS expression. We describe a child with complement-mediated aHUS on maintenance therapy after acute pre-B lymphoblastic leukemia (ALL).

## 2. Case Presentation

A 3-year girl was treated for ALL (pre-B ALL immunophenotype, medium-risk group) according to the treatment protocol ALL IC-BFM 2009. The girl achieved early remission, with negative minimal residual disease (MRD) on day 33 and <1% blasts in bone marrow aspirate. Eight months into ALL maintenance therapy (methotrexate 10 mg/weekly/37 weeks with 6-mercaptopurine and trimethoprim-sulfamethoxazole for infection prophylaxis), during routine follow-up, we observed a sudden onset of pallor, oliguria, and microhematuria with decreased creatinine clearance (Schwartz formula), 23 mL/min/1.73 m^2^. Bone marrow biopsy revealed 2% blasts, with normal erythro- and thrombocytopoiesis. Repeated peripheral blood smears were negative for blasts or schistocytes. Immunohematology analysis (direct and indirect Coombs tests, anti-platelet/erythrocyte antibodies) were negative. Evidence of infective causes was not found (O157:H7, *Shigella* sp., VTEC, Streptococcus pneumoniae). Tests were negative for endomysial antibodies (EMA), ANCA, methylmalonic aciduria, and hyperhomocysteinemia. Plasma-free hemoglobin (PLHBB) was elevated alongside lactate dehydrogenase (LDH), as well as reduced platelet, red blood cell (RBC), and reticulocyte count. ADAMTS-13 activity was reduced (40%, reference range 67–150%) with decreased C3 and normal C4 (Figure 1). Factor H level was elevated (1248 mg/L, ref. range 250–880 mg/L), and terminal pathway activation marker level markedly increased (1315 ng/mL, ref. range 110–252 ng/mL), supporting pathological overactivation of the complement system without the consumption of complement factors. Anti-beta-2-glycoprotein antibodies were slightly elevated (31.3 CU n.v. < 20) with negative anti-factor H IgG autoantibodies, as well as lupus anticoagulant and anti-cardiolipin antibodies [15]. Bilirubin, AST, ALT, GGT, creatinine, d-dimers, fibrinogen, and antithrombin III were elevated before eculizumab treatment and normalized shortly after. Haptoglobin (<0.008 mg/dL) was decreased and normalized after four weeks of treatment. PT and APTT were within reference values (Figure 1). According to the ISTH diagnostic scoring system for DIC guidelines, all values were tested and found negative (<5); 24-h of urine collection showed proteinuria (3.13 g/L), which promptly normalized after eculizumab administration, with a reduction of hemolysis and without clinical signs of nephrotic syndrome. Erythropoietin was elevated (>1500 IU/L). Vitamin B12 and folic acid were within reference ranges. The follow-up of clinical and laboratory parameters are shown in Figure 1. Renal histology was not performed due to negative parental consent regarding the kidney biopsy and the child’s rapid clinical improvement. Appendix A shows diagnostic items and differentials of aHUS in patients with ALL.

All patient follow-up parameters were also analyzed using a correlation matrix (TIBCO Statistica version 12.5) [16]. RPI and RPI/A was calculated from the following formula: RPI = (reticulocyte (%) × patient hematocrit/normal hematocrit)/maturation time. RPI/A has the same formula, only adjusted for median reticulocyte maturation time [17,18]. All reticulocyte counts (RPI, RPI/A, reticulocytes per million, absolute number of reticulocytes) were analyzed in correlation with the platelet count to validate the reticulocyte-platelets relationship. The best correlations were additionally analyzed using Spearman’s rank correlation coefficient (Appendix A). Strong positive correlations were observed with RPI/A vs. platelets, creatinine vs. LDH, and APRI vs. PLHBB variables, while platelets vs. d-dimers, RPI/A vs. serum creatinine, and APRI vs. RPI/A show a strong negative correlation. A positive correlation means that as one parameter increases, the other parameter also tends to increase. A negative correlation signifies that as one parameter increases, the other tends to decrease. The most appropriate correlation was illustrated in GraphPad Prism version 8.4.3.686 and is presented as scatterplots in Figure 2.

The aspartate aminotransferase-to-platelet ratio index (APRI) was calculated from the following formula: ((AST/upper limit of the normal AST range) × 100)/platelet count and matched with RPI/A, d-dimers, AST, and platelets to reveal their potential relationship [19]. RPI/A and platelets both rise with the decrease of hemolysis. It was shown that d-dimers and APRI closely match each other as markers of hemolysis. As the rise of platelets in peripheral blood signifies TMA/aHUS recovery, we observed an inversion point where the decrease of APRI/d-dimers matches the increase of RPI/A and platelets. The rise of RPI/A matches the fall of the APRI at day 9. The graph with statistical analysis was generated in GraphPad Prism version 8.4.3.686, as shown in Figure 3.

## 3. Treatment

The girl was initially treated with fresh frozen plasma, periodic RBC transfusions, and single plasmapheresis. Elevated liver enzymes (AST, ALT) and low urine output indicated the need for plasmapheresis. We immediately switched the maintenance therapy with 6-mercaptopurine to cyclophosphamide. The rest of the maintenance therapy was continued. The child received a first eculizumab application on the second day after admission (300 mg). After administering eculizumab and 6-mercaptopurine discontinuation, PLHBB, LDH, D-dimers, fibrinogen, bilirubin, AST, and ALT decreased rapidly with the rise of platelets, RPI/A and reticulocytes were noticed, which gave us a sign of aHUS improvement. Slightly elevated antithrombin III was normalized after eculizumab administration. A slow but steady global renal function recovery ensued. Total plasma hemoglobin slow improvement is already observed in aHUS recovery.

Moderate hypertension was treated with antihypertensives (enalapril, amlodipine). We believe that excluding 6-mercaptopurine unaccompanied by eculizumab will probably prolong the duration of aHUS with further renal damage. Maintenance treatment with cyclophosphamide was discontinued after 16 months due to negative blasts in bone marrow biopsy and peripheral blood smear. The girl was able-bodied to continue ALL therapy, which is essential for possible ALL relapse. We continued Eculizumab treatment on a recommended schedule for 8 months and decided to cease its administration. After 6 months of follow-up post-eculizumab cessation, we did not find any evidence of hemolysis, ALL relapse, or renal damage.

Genetic analysis: the patient was found to be homozygous for the CFH H3 haplotype (involving the rare alleles of c.-331C > T, Q672Q, and E936D polymorphisms) reported as a risk factor of aHUS, interpreted by geneticists as of uncertain clinical significance. The patient was also homozygous for the MCPggaac haplotype of the CD46 gene reported as a risk factor for developing aHUS. The girl was born to nonconsanguineous parents.

## 4. Discussion and Conclusions

We conducted a systematic review of the literature according to MEDLINE, EMBASE, and Web of Science (WoS) databases. All studies included are selected from December 2021 to the oldest known. The search terms used were “(Leukemia AND aHUS) AND (TMA OR MAHA OR 6-mercaptopurine)”. Inclusion criteria were patients under 18 years old with diagnoses as mentioned above. The EndNote v.20 removed all duplicate references. There were no language restrictions whatsoever. Two researchers (DT, DM) independently selected the articles by title and abstract. All discrepancies were identified and resolved by consensus or with a third investigator (EB).

According to the reviewed literature, the joint association of leukemia and TMA has several possibilities. The first option is that TMA occurs before leukemia, the second concomitantly with leukemia, the third that TMA occurs after leukemia with/without maintenance therapy, and the fourth that it affects leukemia transplant patients.

A total of 312 articles were identified, of which 14 cases were selected for final analysis. In five children, leukemia occurred after a significant period after the resolution of HUS/aHUS [5,6,7,8,14]. In two patients, aHUS occurrence was triggered simultaneously alongside pre-B lymphoblastic leukemia [9,10]. In seven cases (our case included), HUS/aHUS occurred after successful cytostatic therapy following various ALL protocols when the children were on long-term maintenance therapy and the first with clear documentation of eculizumab application [3,10,11,12,13,20]. A triggering factor was most likely drug-induced (6-mercaptopurine) with complement activation amplifying the cycle on a predisposing genetic background. To our best knowledge, this is the seventh well-documented pre-B ALL pediatric patient complicated with aHUS on maintenance therapy and the first to receive Eculizumab regularly.

The lack of such reports poses considerable limitations to our conclusions. We feel that transplant patients have *sui generis* separate aHUS triggers regardless of their underlying diseases and were therefore excluded for further analysis.

A possible conclusion regarding associations between ALL and TMA can be interpreted as that both share a rare common genetic background, or that these two diseases are just coincidental in onset. Most of the authors who encountered these two diseases opted for the first decision. However, antimetabolite drugs used in ALL maintenance therapy might be a separate risk factor for aHUS in children, different from ALL, which could also be an additional independent risk factor. We believe that these three out of seven aforementioned children (our case included) with clear documentation of aHUS can be placed into a particular group. All authors agree that 6-mercaptopurine, save other maintenance drugs, is responsible for the onset of aHUS.

There is a reasonable suspicion that one patient described as HUS and the second with recurrent episodes of HUS had, in fact, aHUS [12,13]. Regardless of the genetic background of the underlying ALL, 6-mercaptopurine appears to be a separate trigger, and these three well-documented cases can also be classified as drug-induced triggers of aHUS. Discontinuation of 6-mercaptopurine from maintenance therapy, alongside supportive therapy, also showed a favorable outcome of aHUS [3]. In conclusion, the exclusion of 6-mercaptopurine is a *conditio sine qua non* in control of such drug-mediated aHUS.

Eculizumab blocked terminal complement activation (C5a and C5b-9), thus reducing RBC hemolysis and platelet consumption. Low hemoglobin levels induced tissue hypoxia, with an increase in erythropoietin plasma levels leading to stimulation of megakaryocytic-erythroid progenitor cells [21]. As platelets and reticulocytes share a common ancestor (bipotent megakaryocytic-erythroid progenitor cell), their levels steadily increased [22]. This shows a close relationship between the changes in hemolysis/erythropoiesis and platelet consumption/megakaryocytopoiesis in favor of cell proliferation (Figure 2, Appendix A). This interval is consistent with the reticulocytosis in peripheral blood and an increase in hematocrit after the slowdown/cease of hemolysis. However, these new RBCs are predominantly reticulocytes with reduced hemoglobin concentration. Consequently, hemoglobin concentration recovery is slower than other monitoring parameters. This observation is supported by elevated erythropoietin and sufficient vitamin B12 and folic acid levels. The corresponding platelets-reticulocytes relationship probably exists in all TMA recovery states, but reticulocyte count was rarely and inconsistently followed in aHUS patients and without comprehensive evaluation. The increase in platelets, RPI, and RPI/A are:Markers of bone marrow response to anemia;Primarily serving as a marker of RBC and platelet recovery;Signifying a favorable response to treatment.

In pediatrics, the APRI was initially implemented in gastroenterology to predict liver injury fibrosis, cirrhosis, and biliary atresia [23,24,25,26,27]. It was applied as a predictor score for carotid intima-media thickness in non-alcoholic fatty liver disease (NAFLD) and, recently, cardiovascular risk in metabolic subjects [28,29]. The index also showed use in follow-up for measuring HIV-related hepatotoxicity and as a prognostic disease risk marker in patients with malaria [30,31]. In liver cirrhosis, AST is elevated due to liver damage, while in HUS/aHUS, elevated AST levels are due to hemolysis, as AST can be found in RBC [32]. Lower levels of ADAMTS-13 result in an inability to cleave ultra-large multimers of the von Willebrand factor resulting in platelet aggregation [33]. After complement-mediated hemolysis and platelet aggregation subside, their levels return to normal, and APRI levels are closer to 0. Since d-dimers are also a parameter of TMA, they are correlated together with APRI for a better assessment of TMA risk [34]. An inverse relationship of APRI/d-dimers and RPI/A along platelets with the *crux* point could be helpful to mark the exact recovery point of aHUS patients and a prognostic marker of eculizumab treatment success.

Because genetic analysis is time-consuming and unavailable during the acute aHUS occurrence, the clinician must initially rely on clinical and laboratory assessment in its decision for eculizumab application. Rapid response to eculizumab, which was observed in our child, may indicate a genetic background prone for its application. At least a short course of eculizumab also benefits the positive course of the disease due to the potential positive effect of crosstalk of coagulation and complement cascade, especially if accompanied by complement–coagulation involvement [35]. Otherwise, genetically inappropriate TMAs for eculizumab application will show limited/no impact on aHUS.

Eculizumab treatment in inflammatory bowel disease, ulcerative colitis, and HELLP syndrome combined with aHUS was already made [36,37,38]. However, the use of eculizumab in aHUS occurrence during ALL maintenance treatment remains controversial. Two possible points should be considered. First, successful eculizumab treatment was essential in controlling the overactivation of the complement system. Homozygosity of CFH H3 haplotype with rare alleles of c.-331C > T, Q672Q, and E936D polymorphisms as well as MCPggaac haplotype of the CD46 gene indicates aHUS genetic background. Steady antithrombin III serum level > 70 and plasma levels of bilirubin, AST, ALT, GGT, and LDH before eculizumab treatment are consistent with aHUS, contrary to the secondary TMA [39].

Secondly, we must consider the possibility of drug-mediated secondary HUS, which in most cases resolves spontaneously after the removal of the underlying cause. However, eculizumab is sometimes also effective in secondary HUS. The possibility of secondary HUS triggering by 6-mercaptopurine was one of the main reasons for eculizumab discontinuation. The second argument for eculizumab discontinuation was the uncertain clinical significance of homozygous CFH H3 haplotype c.-331C > T, Q672Q alleles, and E936D polymorphisms genetic background. In both cases, complement overactivation and amplification removal were crucial for preventing prolonged renal damage. We believe that the limited use of eculizumab in both cases helps alleviate/eliminate the overactivation/amplification of hemolysis. We considered cessation of eculizumab after a short course of treatment [34].

As the girl developed abbreviated laboratory signs of recurrence after 2 months of follow-up (increased PLHBB two times during regular follow-up), we decided to continue complement blockade. This recurrence of hemolysis is attributed to occasional insufficient therapeutic control of aHUS with eculizumab. Therefore, we decided to continue eculizumab use for at least 8 months. Carefully monitoring clinical and hematologic signs of hemolysis, as well as signs and manifestations of kidney damage, we continue to check the girl for the possible hemolytic uremic syndrome and ALL relapse. If another onset of hemolysis occurs, we will consider reintroducing eculizumab or ravulizumab, a more stable permanent treatment for managing aHUS, into continuous therapy.

Considering the length of eculizumab administration, this depends on the clinical and laboratory monitoring alongside the genetic background of each patient. The unknown genetic background for aHUS complicates such a decision. We believe that a determination of the genetic background of aHUS is strongly recommended in cases similar to ours. If genetic mutations pose a risk of aHUS recurrence, the extended use of eculizumab should be considered. According to genetic background, long-term eculizumab/ravulizumab treatment will certainly prevent “silent” renal deterioration. So far, there have been no side effects, hematuria, or deterioration of renal function, which we consider a positive sign for the successful discontinuation of eculizumab treatment. Encouraging is that the use of cyclophosphamide alongside eculizumab did not lead to the relapse of the underlying hematological disease (pre-B ALL) or recurrence of aHUS during follow-up. Therefore, we hope to avoid the “crawling” gradual development of renal dysfunction by aHUS as it may occur in its non-application or premature termination. In doubt, we will again ask permission for a kidney biopsy from the parents.

Previous experiences with ALL and hemolytic uremic syndrome on ALL maintenance therapy are limited. One child with pre-B ALL had a spontaneous remission of aHUS after discontinuation of 6-mercaptopurine speaks against eculizumab routine administration [3]. Slow renal function recovery and residual hypertension were observed in two children, and long-term renal impairment should be considered in these children [3]. If not treated properly, residual/progressive renal impairment remains open in cases similar to ours. We did not find convincing evidence of eculizumab use in a child with aHUS who eventually developed chronic kidney disease. Accordingly, we must assume that eculizumab/ravulizumab administration, in addition to the genetic support for its need, is beneficial for maintaining long-term renal function. Unfortunately, we do not have the genetic background of this child [3].

The genetic background of leukemia and aHUS may be related, triggering one another. Therefore, a weak genetic connection of both diseases may be considered. To our best knowledge, this assumption has limited support in the current scientific literature [3,4,5,7,9,10,12,13]. Judging by the total number of ALL patients on chemotherapy globally, the likely explanation so far is that these diseases are genetically independent and associated only by coincidental genetic matching for both disorders.

We suspect two insufficiently documented additional cases with the use of 6-mercaptopurine as undiagnosed aHUS because clinical and laboratory data strongly suggests this [12,13]. There is also a strong suspicion that one child diagnosed as HUS and successfully treated with eculizumab was indeed aHUS [14]. Therefore, we may assume that the two diseases could be under-or misdiagnosed and later became evident by late renal damage, which is then erroneously classified as chemotherapy-related. Some of TMA/ALL occurrences may be underreported as well. Close monitoring of hemolysis in children during chemo- and maintenance therapy and post bone marrow transplanted patients may reveal the correct number of these two diseases associations. Complement monitoring should be reserved for selected cases. If complement activation and consumption with terminal complement complex C5b-9 are detected, a short course/long-term eculizumab/ravulizumab administration should be considered [34]. In such cases, genetic analysis for aHUS is recommended.

## Figures and Tables

**Figure 1 jcm-11-02779-f001:**
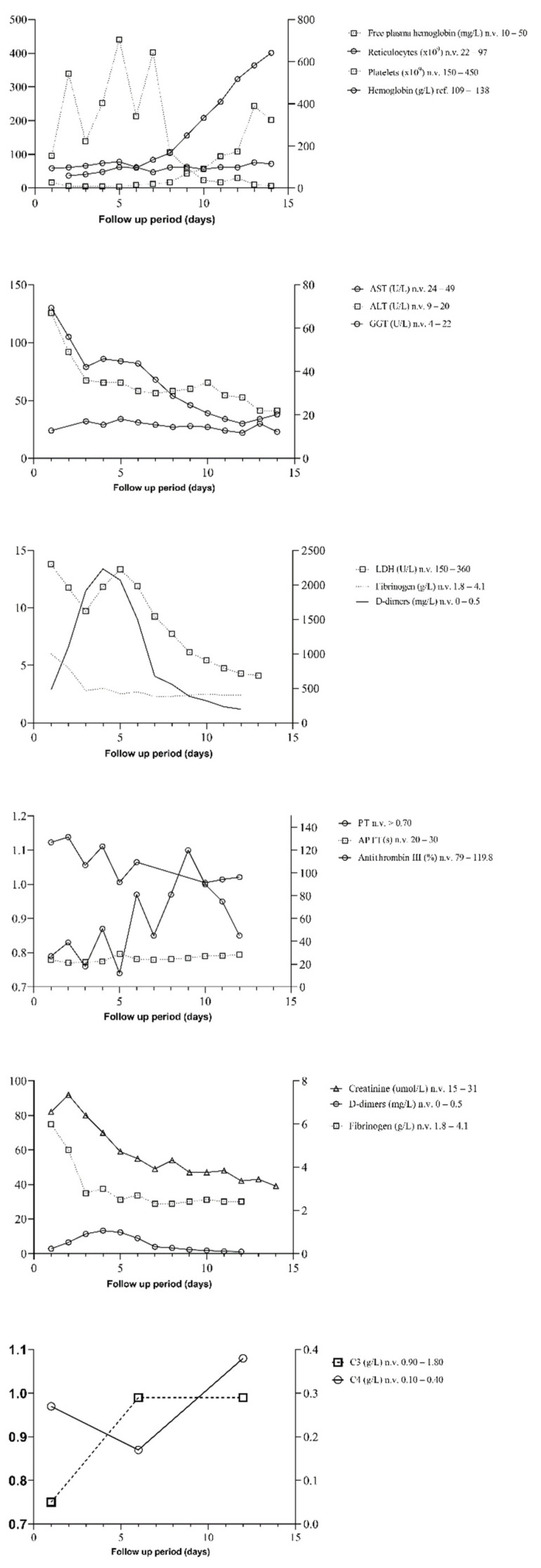
Clinical, laboratory, and treatment time-line follow-up. Abbreviations: AST = aspartate aminotransferase; ALT = alanine transaminase, GGT = Gamma-glutamyltransferase, LDH = Lactate dehydrogenase, PT = prothrombin time, APTT = activated partial thromboplastin time.

**Figure 2 jcm-11-02779-f002:**
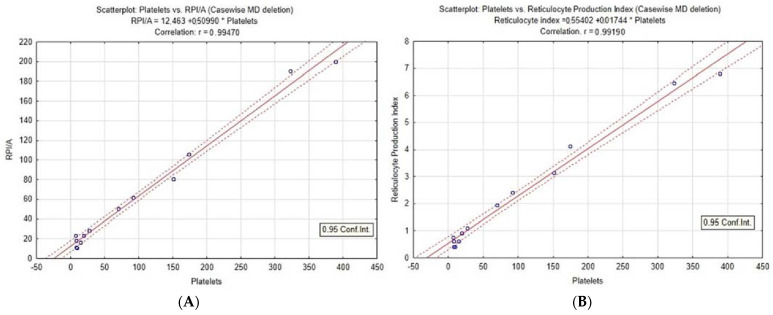
Correlation of platelets with RPI (**A**) and RPI/A (**B**), Spearman rank correlation coefficient *p* < 0.001, R > 0.99. Reticulocyte index with age correction shows the best correlation. Both RPI (Reticulocyte Production Index) and RPI/A (Reticulocyte Production Index adjusted for age) show a high correlation with platelets.

**Figure 3 jcm-11-02779-f003:**
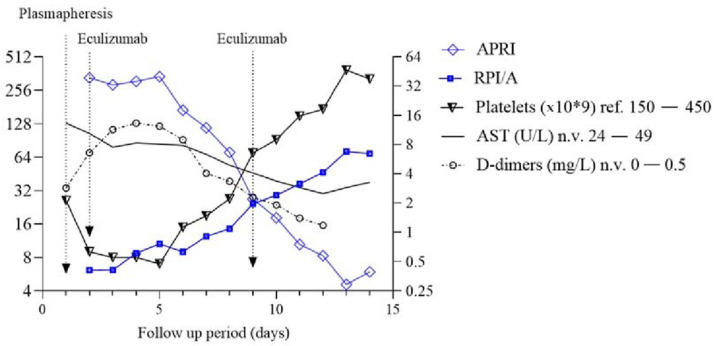
Correlation of aspartate aminotransferase-to-platelet ratio index (APRI) with d-dimers, RPI/A (Reticulocyte Production Index adjusted for age) and platelet count. The APRI score closely monitors d-dimer levels (Spearman test, *p* = 0.018, R = 0.923) and RPI/A (Spearman test, *p* < 0.001, R = −0.94). Approximately 9 days after the initial symptoms, an inverse manner following the *crux* of APRI/platelet values with d-dimers and RPI/A is observed. APRI and RPI/A variables are highlighted. The figure is presented in log2 fashion on the Y axis to better illustrate convergence of their variables. AST = aspartate aminotransferase.

## Data Availability

Data available upon request.

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
