# Peer review of "A Limited Course of Eculizumab in a Child with the Atypical Hemolytic Uremic Syndrome and Pre-B Acute Lymphoblastic Leukemia on Maintenance Therapy: Case Report and Literature Review"

_jcm, 2022, doi:10.3390/jcm11102779_

Round 1

Reviewer 1 Report

Dear authors

You described the case of a girl with a secondary form of aHUS related to pre-B ALL maintenance treatment associated with two high risk polymorphisms in homozygosis  (CFH H3 and MCPggaac), who was treated successfully with eculizumab. The patient received eculizumab for 8 months without relapse.

This case illustrates the clinical picture of aHUS in patients with ALL,its differential diagnosis, and the relevance of early specific treatment.

However, before considering it for publication , I suggest some changes , as described below:

1) a short introduction about main diagnostic items and differentials of aHUS in patients with ALL would be of interest (at least as supplementary material)

2) you suggested the use of two parameters:  Production Index (RPI) to platelet ratio and aspartate-aminotransferase-platelet ratio index (APRI), as markers of bone marrow response to eculizumab, as red blood cells and platelets response respectively , as a way to measure the response to eculizumab

Despite the idea is interesting it would be nice to further discuss the relationship with other parameters as LDH, serum creatinine, high blood pressure, protein /Creatinine ratio in urine… it is hard to support that hypothesis based only on one patient. What is in your opinion the reason that those indicators of bone marrow response are not part of the standard of care? Are they specific of patients with hematologic disorders? Are they used in other patients with hematologic diseases or other types of TMA or HUS?

It is recommended to further discuss the role of those indexes at the discussion section

3) other comments

The number of figures is excessive. Please consider moving some of them to a supplementary section

Author Response

Dear Reviewer #1,

Thank you for your kind suggestions. We will answer your questions according to your points.

  • A short introduction about main diagnostic items and differentials of aHUS in patients with ALL would be of interest (at least as supplementary material)

Ad#1: To make a simplified approach, we created a diagram to illustrate the main items and differences in patients with aHUS and ALL. I hope that you will find it informative. According to your suggestions, it was placed in the supplementary material (Supplementary diagram 1).

  • Despite the idea is interesting it would be nice to further discuss the relationship with other parameters as LDH, serum creatinine, high blood pressure, protein /Creatinine ratio in urine… it is hard to support that hypothesis based only on one patient. What is in your opinion the reason that those indicators of bone marrow response are not part of the standard of care? Are they specific of patients with hematologic disorders? Are they used in other patients with hematologic diseases or other types of TMA or HUS?

Ad#2: Thank you for your suggestion to correlate different variables. We constructed a correlation matrix with a Spearman rank correlation coefficient to better present the relationship between the aforementioned parameters (LDH, serum creatinine, high blood pressure, protein/creatinine ratio in urine). Variables with p<0.01 were considered significant and added in Supplementary Table 1. Strong positive correlations were observed with RPI/A vs. platelets, creatinine vs. LDH, and APRI vs. PLHBB variables, while platelets vs. d-dimers, RPI/A vs. serum creatinine, and APRI vs. RPI/A show a strong negative correlation. A positive correlation means that as one parameter increases, the other parameter also tends to increase. A negative correlation signifies that the other parameter tends to decrease as one parameter increases.

Both pediatric hematologists and nephrologists did monitoring of the patient. Since pediatric hematologists were concerned with bone marrow recovery after ALL-BFM 2009 treatment protocol, we integrated reticulocytes into our follow-up, alongside other CBC parameters. Elevated reticulocyte levels have been noticed sporadically in hemolytic anemias (HUS, aHUS, TMAs) due to hemolysis, but no systematic statistical analysis and comparison with other parameteres was attempted. Since d-dimers are also a parameter of TMA, they are correlated together with APRI for a better assessment of TMA risk. The corresponding platelets-reticulocytes relationship probably exists in all TMA recovery states, but reticulocyte count was rarely and inconsistently followed in aHUS patients and without comprehensive evaluation. We found the use of APRI score in one patient with HELLP syndrome (TMA).

  • It is recommended to further discuss the role of those indexes at the discussion section

Ad#3 We added the formula for reticulocyte production index (RPI). RPI = (reticulocyte (%) × patient hematocrit / normal hematocrit) / maturation time which was missing in the text. The RPI is a good indicator of the bone marrow erythropoietic response to anemia. It is a marker of RBC recovery and signifies a favorable response to treatment. The corrected reticulocyte production index RPI/A has better values than RPI as it is tailored to the maturation time of reticulocytes. The origin of the relationship between reticulocytes and platelets was also addressed, as both share a similar megakaryocytic-erythroid progenitor. This progenitor cell is stimulated by tissue hypoxia, which increases erythropoietin plasma levels.

APRI score is calculated from the following formula: AST x AST upper limit x 100 divided by platelets. In liver cirrhosis, AST is elevated due to liver damage, while in HUS/aHUS, elevated AST levels are due to hemolysis, as AST can be found in red blood cells. In HUS/aHUS patients RBC and platelet levels are reduced because of complement-mediated hemolysis of RBC and aggregation of platelets. After cessation of hemolysis and platelet aggregation, their levels return to normal, and APRI score levels return closer to 0. Since d-dimers are also a parameter of TMA, they are correlated together with the APRI index for a better assessment of TMA risk.

  • According to your suggestions Figure 3 was moved as supplementary material (Supplement figure 1).

  • We added in Figure 3 new variables such as AST and RPI/A. The figure is presented in log2 fashion on the Y-axis to better illustrate the convergence of their variables. We can either replace the old figure with the new figure or leave both figures in the text. Please let us know which of these two options is more appropriate for you?

  • Due to your stimulative comments, we did an additional analysis with the Spearman rank correlation coefficient between APRI with RPI. The result is p<0.01, and rs is -0.94 (Supplementary Table 1, Figure 3).

Due to word count, we also reduced the abstract to less than 200 words.

We believe that the comments from the reviewers have substantially improved our manuscript. We look forward to your decision on whether this improved version would be considered acceptable for publication in your journal.

Looking forward to hearing from you soon,

Daniel Turudic

Reviewer 2 Report

Dr Turudic and co-authors have done a commendable job of presenting a case of aHUS in a pre-B ALL patient that responded to Eculizumab therapy. They have additionally done a literature search on the same topic and have mentioned 7 cases of aHUS and ALL.  Additionally they have tried to show value of usage of Reticulocyte Production Index (RPI) to platelet ratio and and aspartate-aminotransferase-platelet ratio index (APRI) in treatment with Eculizumab therapy. 

Well written case report and discussion of the case and literature

Major concerns: 

  1. Use of these new hematological indices: RPI to platelet ratio - it would be helpful if you can justify the use of this index and how the correlation exits between reticulocyte count index and platelet count in aHUS. Is the reticulocyte low initially in aHUS because of the patient being very sick and there is bone marrow suppression and on administration of Eculizumab, not only is there reduction of platelet destruction but also significant increase in reticulocyte production - kindly explain
  2. APRI to platelet ratio and d-dimer levels - from your figure, d-dimer levels and APRI levels started to drop significantly from day 6 onwards despite the platelet counts being low then - hard to justify that day 10 is the peak or crucial point in improvement after Eculizumab therapy. Kindly explain
  3. Discussion - Line 174: All authors agree that 6-mercaptopurine, save other maintenance drugs, is responsible for the onset of aHUS - is this not contradictory to what you write later in your discussion - the patient required re-initiation of Eculizumab therapy as there recurrence of aHUS features despite stopping 6-mercaptopurine. 
  4. Minor issues: a)Abstract Line 20-26: need grammatical correction. b) case report: Line 61: We found in the literature that reticulo-61 cyte count was rarely and inconsistently followed in aHUS patients and without comprehensive evaluation - kindly delete or lese move it to discussion area. c) Case report: line 72: PV and APTV were within ref- erence values (Figure 1).- I think you mean PT and APTT. d) Line 215: Rapid response to Eculizumab which was observed in our child may indicate a favorite genetic background for its application (kindly provide reference)

Author Response

Dear Reviewer #2,

Thank you for you for your kind review, we had a similar question from other reviewers.  We hope that we adressed your concerns below.

  • Use of these new hematological indices: RPI to platelet ratio - it would be helpful if you can justify the use of this index and how the correlation exits between reticulocyte count index and platelet count in aHUS. Is the reticulocyte low initially in aHUS because of the patient being very sick and there is bone marrow suppression and on administration of Eculizumab, not only is there reduction of platelet destruction but also significant increase in reticulocyte production - kindly explain

Ad#1 You are correct. We did not explain our formulas and indices very well nor how we calculated them. Therefore we added in the discussion section the following sentences "RPI and RPI/A is calculated from the following formula: RPI = (reticulocyte (%) × patient hematocrit / normal hematocrit) / maturation time. RPI/A has the same formula, only adjusted for median reticulocyte maturation time. "RPI/A and platelets both rise with the cessation of hemolysis and can be a suitable correlation matrix for TMA recovery. The best correlations were additionally analyzed using Spearman's rank correlation coefficient (Supplementary Table 1). Strong positive correlations were observed with RPI/A vs. platelets, creatinine vs. LDH, and APRI vs. PLHBB variables, while platelets vs. d-dimers, RPI/A vs. serum creatinine, and APRI vs. RPI/A show a strong negative correlation. A positive correlation means that as one parameter increases, the other also tends to increase. A negative correlation signifies that the other tends to decrease as one parameter increases. The RPI/A was initially low because of strong hemolysis, while bone marrow RBC and reticulocyte production could not cope with the RBC destruction rate.

There have been no reports of Eculizumab inducing bone marrow suppression or failure published yet. Rapid bone marrow recovery after the first 5 days after Eculizumab administration confirms no bone marrow suppression. Eculizumab blocked terminal complement activation (C5a and C5b-9), thus reducing RBC hemolysis and platelet consumption. Low hemoglobin levels induced tissue hypoxia, with an increase in erythropoietin plasma levels leading to stimulation of megakaryocytic-erythroid progenitor cells. As platelets and reticulocytes share a common ancestor (bipotent megakaryocytic-erythroid progenitor cell), their levels steadily increased. This shows a close relationship between the changes in hemolysis/erythropoiesis and platelet  consumption/megakaryocytopoiesis in favor of cell proliferation (Figure 2, Supplementary figure 1).

  • APRI to platelet ratio and d-dimer levels - from your figure, d-dimer levels and APRI levels started to drop significantly from day 6 onwards despite the platelet counts being low then - hard to justify that day 10 is the peak or crucial point in improvement after Eculizumab therapy.

Ad#2  D-dimers and APRI index did fall significantly from day 6 onwards, but platelet count also showed considerable improvement too. From the new Figure 3 it is shown that RPI/A matches the same point as APRI and D-dimers on the same day 9. Therefore, all three variables converge at the same point.

AST is commonly found in the heart, liver, skeletal muscle, and red blood cells. As AST levels are elevated in hemolysis, their gradual decrease is observed 4 days after Eculizumab administration. The relationship between AST and platelets changes inversely on day 6 after Eculizumab administration as the platelet numbers rise. The change in ratio is due to a decrease in AST and an increase in platelet levels. D-dimers continue to drop as the platelet levels increase. This negative correlation is described in Figure 3 and Supplementary Table 1. The increase in platelet levels is significantly higher than the decrease in d-dimers. Therefore we added in Figure 3 new variables such as AST and RPI/A. The figure is presented in log2 fashion on the Y-axis to better illustrate the convergence of their variables. We can either replace the old figure with the new figure or leave both figures in the text. Please let us know which of these two options is more appropriate for you?

  • Discussion - Line 174: All authors agree that 6-mercaptopurine, save other maintenance drugs, is responsible for the onset of aHUS - is this not contradictory to what you write later in your discussion - the patient required re-initiation of Eculizumab therapy as there recurrence of aHUS features despite stopping 6-mercaptopurine.

Ad#3. I believe that we did not express our presentation clearly. The patient was on Eculizumab continuously for 8 months in consultation with a hematologist, nephrologist and geneticist who did not know whether the mutation was pathogenic or not. We considered cessation of Eculizumab after 2 months, but after the signs of hemolysis, we decided to continue Eculizumab treatment. We believe that removal of 6-mercaptopurine only removed the trigger of aHUS, but the downregulation of regulators was prolonged further. We discontinued Eculizumab only after 8 months of treatment but will return to the first sign of hemolysis or transfer to Ravulizumab due to homozygosity of CFH H3 haplotype (involving the rare alleles of c.-331C>T, Q672Q, and E936D polymorphisms of uncertain significance).

  • Abstract Line 20-26: need grammatical correction. b) case report: Line 61: We found in the literature that reticulo-61 cyte count was rarely and inconsistently followed in aHUS patients and without comprehensive evaluation - kindly delete or lese move it to discussion area. c) Case report: line 72: PV and APTV were within ref- erence values (Figure 1).- I think you mean PT and APTT. d) Line 215: Rapid response to Eculizumab which was observed in our child may indicate a favorite genetic background for its application (kindly provide reference)

Ad#4 Lines 20-26 are corrected as the abstract was reduced due to word count. Line 61 was relocated to the Discussion section. In Line 72 PV and APTV were changed to PT and APTT.

The meaning of Line 215 was poorly written, so we changed the structure of the sentence. It was meant that if hemolysis again occurs, the patient, after all has a genetic profile susceptible to aHUS treatment with Eculizumab/Ravulizumab.  

  • Due to your stimulative comments, we did an additional analysis with the Spearman rank correlation coefficient between APRI with RPI. The result is p<0.01, and rs is -0.94 (Supplementary Table 1, Figure 3). We constructed a correlation matrix with a Spearman rank correlation coefficient to better present the relationship between the aforementioned parameters (LDH, serum creatinine, high blood pressure, protein/creatinine ratio in urine). Variables with p<0.001 were considered significant and added in Supplementary Table 1.

Due to word count, we also reduced the abstract to less than 200 words.

We believe that the comments from the reviewers have substantially improved our manuscript. We look forward to your decision on whether this improved version would be considered acceptable for publication in your journal.

Looking forward to hearing from you soon,

Daniel Turudic